# Versatile Cell and Animal Models for Advanced Investigation of Lead Poisoning

**DOI:** 10.3390/bios11100371

**Published:** 2021-10-04

**Authors:** De-Ming Yang, Yu-Fen Chang

**Affiliations:** 1Microscopy Service Laboratory, Basic Research Division, Department of Medical Research, Taipei Veterans General Hospital, Taipei 11217, Taiwan; 2Institute of Biophotonics, School of Biomedical Science and Engineering National Yang Ming Chiao Tung University, Taipei 11221, Taiwan; 3National Biotechnology Research Park, LumiSTAR Biotechnology Inc., Taipei City 115, Taiwan; yang.deming2021@nycu.edu.tw

**Keywords:** *Drosophila melanogaster*, fluorescence resonance energy transfer, Met-lead 1.44 M1, Pb biosensor

## Abstract

The heavy metal, lead (Pb) can irreversibly damage the human nervous system. To help understand Pb-induced damage, we applied a genetically encoded Förster resonance energy transfer (FRET)-based Pb biosensor Met-lead 1.44 M1 to two living systems to monitor the concentration of Pb: induced pluripotent stem cell (iPSC)-derived cardiomyocytes as a semi-tissue platform and *Drosophila melanogaster* fruit flies as an in vivo animal model. Different FRET imaging modalities were used to obtain FRET signals, which represented the presence of Pb in the tested samples in different spatial dimensions. Using iPSC-derived cardiomyocytes, the relationship between beating activity (20–24 beats per minute, bpm) determined from the fluctuation of fluorescent signals and the concentrations of Pb represented by the FRET emission ratio values of Met-lead 1.44 M1 was revealed from simultaneous measurements. Pb (50 μM) affected the beating activity of cardiomyocytes, whereas two drugs that stop the entry of Pb differentially affected this beating activity: verapamil (2 μM) did not reverse the cessation of beating, whereas 2-APB (50 μM) partially restored this activity (16 bpm). The results clearly demonstrate the potential of this biosensor system as an anti-Pb drug screening application. In the *Drosophila* model, Pb was detected within the adult brain or larval central nervous system (Cha-gal4 > UAS-Met-lead 1.44 M1) using fast epifluorescence and high-resolution two-photon 3D FRET ratio image systems. The tissue-specific expression of Pb biosensors provides an excellent opportunity to explore the possible Pb-specific populations within living organisms. We believe that this integrated Pb biosensor system can be applied to the prevention of Pb poisoning and advanced research on Pb neurotoxicology.

## 1. Introduction

Toxic elements, such as the heavy metal, lead (Pb), have been affecting the daily lives of humans for more than three thousand years. Pb has been used to adulterate wine and gasoline, to make water pipes (since the ancient Roman period), in traditional Chinese medicine, and as a component of glass and ceramics, paint, hair dyes, children’s toys, and batteries. The recognition that Pb damages human tissues was only reached after countless sad historical events. For example, during the last Arctic expedition of Sir John Franklin in 1845, all the crew members of the ships Erebus and Terror were found dead two years after they had gone missing. Detailed investigations suggested that in addition to tuberculosis, Pb poisoning from contaminated tinned food and water was an important factor in these deaths [1,2,3,4,5,6]. The neurological effects of high Pb exposure include delusion, impaired judgment, and even a desire to commit murder. An increased Pb content in honey produced near the Cathedral of Notre-Dame de Paris in France after a 2019 fire was found to derive from old leaded stained glass that was destroyed by the fire, producing fumes that were spread by the wind [7,8]. More recently, a doctor of Chinese medicine in Taiwan illegally issued an incorrect prescription containing large amounts of Pb, which led to harmful effects for patients with ages ranging from 4 to 85 years old [9]. Pb can cause tissue damage that leads to neurological disorders, including cognitive deficits, low intelligence quotient (IQ), and attention-deficit hyperactivity disorder (ADHD) [10]. In addition, Pb can cause cardiovascular diseases (CVDs) such as secondary arterial hypertension [11], and circulatory illnesses including anemia [12]. These toxicological impacts of Pb can result from various pathways of Pb exposure, such as drinking contaminated water, breathing, eating, and physical contact with Pb.

Fluorescent indicators can be in chemical form, which are preferentially applied for environmental sensing; alternatively, they can be genetically encoded (GE), as with fluorescent protein (FP) biosensors (i.e., GEFP biosensors) that can be expressed intracellularly to sense specific targets. In combination with suitable readout platforms, both kinds of fluorescent indicators can be applied to monitor the presence of specific compounds or elements within living systems or in the environment [13,14]. Strategies for the qualitative and quantitative sensing of targets by GEFP biosensors can include the combination of a bioreceptor (a selected sensing domain) with a signal transducer, which transforms target recognition events into fluorescent signals. For example, the first GEFP biosensor, cameleon, was developed to detect Förster resonance energy transfer (FRET) signal events between two pairs of FPs (blue fluorescent protein, BFP with green fluorescent protein, GFP, or cyan fluorescent protein, CFP with yellow fluorescent protein, YFP) following the sensing of intracellular calcium ions [15]. This novel strategy makes it possible to visualize specific molecules in real-time and in a genetically controlled manner: the biosynthesis of a FRET-based biosensor proceeds through DNA transfer and replication, mRNA transcription, and protein translation inside living cells containing the biosensor gene without the need to stain tissues [13].

This strategy inspired the development of different types of FRET-based biosensors around the world to examine the dynamic status of living cells and thus better understand the functional roles of molecules of interest, to precisely and rapidly predict the fate of targeted cells, and, in particular, to provide a powerful platform for drug screening [16,17]. As they require high molecular selectivity, however, GEFP biosensors to detect metal ions, such as zinc (Zn^2+^), copper (Cu^2+^), silver (Ag^+^), cadmium (Cd^2+^), and Pb^2+^, have been relatively difficult to generate [18,19,20,21,22,23]. Given the importance of monitoring heavy metals in environmental and biological samples, biosensors that can perform both these functions are especially desirable.

To monitor Pb toxicity, we designed the FRET-based GEFP biosensor Met-lead (Figure 1), which we reported previously [20]. However, the sensing ability of the first version, Met-lead 1.59 (Figure 1A), provided a dynamic range (DR) of less than two-fold and a limit of detection (LOD) of about 500 nM (~10 μg/dL; compared with a recommended maximum blood lead level for children of 2 μg/dL), making it unsuitable for further applications [20]. Therefore, we endeavored to improve the sensitivity of Met-lead molecules in different ways, including adjusting the lengths of the Pb-sensing domains and inserting a repeat sequence (linker) in the center of the Pb-sensing domains [23]. We achieved a LOD of Pb of 10 nM (2 ppb) for the most recent Met-lead version, 1.44 M1 (Figure 1B) under suitable filter setup (Appendix A and Figure 1C,D). This offers great potential for important practical applications such as the detection of Pb in the environment (in water for drinking or irrigation), in body fluids (in cells, serum, or urine), or in other tissues (in vivo in whole animals or plants), filling a need that has been widely noted by researchers over the years. In this study, we used two different systems to demonstrate the measurement of Pb under various conditions: induced pluripotent stem cell (iPSC)-derived cardiomyocytes [24,25] with measurable functional parameters, which provide a unique in vitro tissue model, and the adult brain and larval central nervous system (CNS) of transgenic *Drosophila*, as an easy-to-handle in vivo model [23,26].

## 2. Materials and Methods

### 2.1. Cell Culture and Drug Pretreatments

The human iPSC-derived cardiomyocytes (Human iPSC Cardiomyocytes-Cardiosight^®^-S) [24,25], were obtained from Nexel. A 96-well glass-bottom plate was first coated with 0.5% fibronectin/0.1% gelatin at 37 °C for at least 1 h. The human iPSC cardiomyocytes were transferred into the plate wells and cultured for 3 days in Nexel’s Cardiomyocyte Maintenance Medium, which was then replaced with Tyrode’s buffer. For FRET imaging, the human cells were seeded onto 24-mm cover-glasses coated with poly-L-lysine and then transfected with Met-lead genes using Lipofectamine (Thermo Fisher Scientific) according to the manufacturer’s instructions [20,26,27]. The cells were used for lead biosensing 2 days after transfection [23]. Pb (50 μM) was introduced at the time points indicated by arrows as shown in Figures. Verapamil (final concentration 2 μM) or 2-APB (final concentration 50 μM) were immersed into the buffer containing 2 mM Ca 10 min before the FRET experiments with Pb.

### 2.2. Fly Strains and Transgenic Fly Constructs

The *Drosophila melanogaster* stocks were raised on a standard cornmeal medium and housed at 25 °C with 70% relative humidity in a 12 h:12 h light:dark cycle incubator. To measure the contents of Pb within the cholinergic neurons of adult brains or the larval CNS, the strain *Cha-gal4* was used and was injected with UAS-Met-lead 1.44 M1 [23,26].

### 2.3. Preparation of Fly Samples

For in vivo (rapid) time-lapse FRET biosensing of Pb (10 μM) in the adult fly brain, flies were placed into a custom-made plate and the brain of each fly was removed under the FRET microscope [23]. For the larval CNS, each intact larva was cooled on ice and plated onto the FRET microscope for rapid image acquisition [26].

For in situ or ex vivo (high-resolution, two-photon illumination) FRET biosensing of Pb within the larval CNS, whole larva bodies were immersed with Pb (10 μM) for 3 h before fixated and cleared. The whole larva bodies were in 4% paraformaldehyde pre-cooled on ice and shaken gently for 1 h. The samples were transferred to 2% PBST (PBS with 2% Triton X-100) for 30 min and then shaken and placed in a vacuum chamber. The samples were then mounted onto coverslips and cleared with 15 μL RapiClear (SunJin Lab, Co, Taiwan), and were stored in an electronic dry cabinet until imaging.

### 2.4. FRET-Based Pb Imaging

For in-cell or in vivo FRET ratio imaging of Pb within iPSC-derived cardiomyocytes or in the *Drosophila* brain, an inverted microscope (Axiovert 200 M, Zeiss, Germany) or a stereomicroscope (MVX-10, Olympus, Japan) equipped with a 440 nm light source and a W-View module (Gemini, Hamamatsu, Japan; with filters 542/27 nm for YFP and 483/32 nm for CFP; Appendix A and Figure 1C,D) and a CMOS camera (ORCA-Flash4.0, Hamamatsu, Japan) controlled by HCImage software was used. The fluorescent signals of cp173Venus and ECFP(ΔC11) from samples that expressed Met-lead were rapidly acquired from the in-cell or in vivo FRET Y/C ratiometric imaging systems.

For the high-resolution FRET Y/C ratiometric imaging of larval CNS, a two-photon microscope was used. An 850-nm two-photon laser was applied as an excitation source within a multiphoton microscope (Zeiss LSM 7 MP, with 20×, NA 1.0 water objectives, Germany). The emission signals of YFP (530–630 nm) and CFP (460–500 nm) were acquired separately (Appendix A and Figure 1C,D).

### 2.5. Data Analysis

ImageJ was used to combine the ratio of fluorescent signals of cp173Venus (YFP channel) and ECFP(ΔC11) (CFP channel) using the Ratio Plus plugin and displayed with a rainbow-colored palette using the Lookup Table (LUT) to visualize FRET Y/C ratio biosensing (colored from blue to red according to the lowest to the highest ratio value) within intact living cells, fly brains or the larval CNS.

## 3. Results

### 3.1. Simultaneous Monitoring of Physical Cardiomyocyte Activity and Pb Content

The cardiovascular system is one of the most important targets of Pb toxicity [27,28]. The entry of Pb into the cardiovascular system has been suggested to occur via various calcium (Ca^2+^) channels [29,30], and Pb impairs the normal function of cardiac contraction [27,31,32]. We used the improved Met-lead 1.44 M1 Pb biosensor in iPSC-derived cardiomyocytes to demonstrate the sensing ability of Met-lead (FRET ratio increase; Figure 2F,G; original fluorescent images of Met-lead 1.44 M1 in YFP and CFP channels are shown in Appendix A) while simultaneously obtaining physiological parameters (e.g., the cardiomyocyte beating rate; Figure 2D,E). The beating frequency (beats per minute, bpm) of the cardiomyocytes (20–24 bpm) was significantly affected by exposure to Pb (50 μM; Figure 2F,G; Table 1; beating none detected, NA). Pretreatment with the calcium channel blocker verapamil (2 μM) not only impaired the beating function of cardiomyocytes (Figure 3D and Table 1; original fluorescent images of Met-lead 1.44 M1 in YFP and CFP channels are shown in Appendix A; beating NA), but also decreased the entry of Pb (Figure 3F and Table 1; original fluorescent images of Met-lead 1.44 M1 in YFP and CFP channels are shown in Appendix A). Another calcium channel blocker, 2-aminoethoxydiphenyl borate (2-APB; 50 μM; the store-operated Ca-channel inhibitor), altered the rate of beating (Figure 3E and Table 1; 16 bpm). This experiment provided a time-lapse recording of Met-lead as a classical demonstration of in situ Pb monitoring, and also generated additional information about the health status (beating rate) of living cardiomyocytes.

### 3.2. In Vivo Pb Sensing

The impact of Pb on the nervous system has been noted for years [10,12]. In this study, we introduced the improved Met-lead 1.44 M1 (Figure 1B) biosensor gene into *Drosophila* with tissue-specific expression of the upstream activating sequence controlling the gene’s expression (UAS-GAL4 flies). In this way, the performance of the Pb biosensor within targeted organs at different life stages, such as the adult brain (Figure 4) or larval CNS (Figure 5), could be detected. In adult flies that expressed the Pb biosensor within cholinergic neurons (*Cha-gal4* > *UAS-Met-lead 1.44 M1*, Figure 4A,B), changes in the FRET ratio signal within the neurons of the brain were visualized as rainbow-colored images (Figure 4C) or were displayed as line plots for selected regions (Figure 4D). These increased FRET ratio signals not only represent the presence of Pb in situ (Figure 4C,D) but also provide a direct demonstration that the Pb biosensor can be used to detect the Pb content within living organisms. We further applied a two-photon FRET platform to show the single-cell image quality of the intact adult fly brain ex vivo (Appendix A and related animations in 3D or optical sections) and even in vivo alive (Appendix A and related animations in 3D or optical sections).

We also used two-photon FRET imaging to monitor the concentration of Pb within the intact larval CNS (*Cha-gal4* > *UAS-Met-lead 1.44 M1*, Figure 5) in a 3D manner (Figure 5A) or using sectional imaging (Figure 5B; for representative sections; original fluorescent images of Met-lead 1.44 M1 in YFP and CFP channels are shown in Appendix A). The ratio values within different regions of interest along vertical sections of the larval CNS are shown in Figure 5C.

## 4. Discussion

Although the use and acceptable concentrations of Pb are highly regulated, Pb continues to harm human health and cause neurological disorders [10] and/or CVDs [11], even at relatively low levels that may be under the critical limit allowed in a particular jurisdiction [10,12]. An advanced tool to indicate the presence of Pb in the environment or in human daily inputs such as water, food, and merchandise would be extremely beneficial. In this study, we demonstrate practical applications of the optimized FRET-based genetically encoded Pb biosensor Met-lead 1.44 M1 (Figure 1B) in two different systems: in vitro in iPSC-derived cardiomyocytes (Figure 2 and Figure 3) and in vivo in fruit flies (Figure 4 and Figure 5).

Since crystallography images of PbrR (the key to Pb sensing of Met-leads) are currently unavailable, the current structural knowledge of Met-leads relies on molecular simulation, which still needs to be confirmed. Fortunately, PbrR belongs to the MerR superfamily, and the structures of some MerR members have been resolved by crystallography, e.g., MerR [33]. Homology modeling has been used to display the structures of PbrR [23] and Met-leads (Figure 1A,B) [23,26] using structural information about MerR as a reference [33]. The distance between the donor and acceptor inside Met-lead 1.44 appears to be farther than that inside Met-lead 1.59. Perhaps there is more space between FRET pair proteins inside Met-lead 1.44 compared to Met-lead 1.59, thus generating various types of flexible conformations for FRET and producing wider ranges of FRET ratio values. This could explain why Met-lead 1.44 has higher dynamic ranges and sensitivity. Furthermore, the linker version of Met-lead (1.44 M1) could generate additional space between FRET pair proteins (lower ratio value at resting [23]). Whether this difference in distance is the main reason for the higher FRET efficiency of Met-lead 1.44/1.44 M1 than Met-lead 1.59 remains to be verified using solid crystallography data in the future.

Our results of qualitative analysis of Y/C ratios based on the current experimental data are similar to those of previous reports [15,23,26]. The increased FRET ratio of Met-leads is confirmed to be due to Pb signals for the following reasons: (1) The filter setup (e.g., the W-View module and the setting within two-photon FRET) and the laser illumination were based on spectral information for ECFP, EYFP/Venus, and Met-leads (Appendix A and Figure 1C,D). This setup effectively separates the emission of CFP from that of YFP, thus avoiding the cross-over between CFP and YFP signals. (2) The increased ratio values of Met-leads were previously shown to arise from Pb using the Pb chelator TPEN, which was shown to be similar to the use of EGTA for Ca biosensing. Although such increased Y/C ratio values (under donor/CFP-specific excitation) might be too simple compared to the classical FRET calculations, as the original theory established [34], it is relatively convenient to use the above Y/C ratios for practical applications such as on-site real-time quantitative environmental detection of Pb. In the future, by combining the titration of Pb at various concentrations (e.g., from 1 nM to 50 μM) [23] with a newly developed portable smartphone device [35], rapid Pb detection will be available, preventing the risk of Pb poisoning. Thus, information about acceptor/YFP-excited signals might not be essential for this type of biosensing.

As concerns exist about the clinical use of chelating agents such as dimercaprol (British anti-Lewisite, BAL) and calcium disodium ethylenediaminetetraacetic acid (CaEDTA) to treat Pb poisoning [36,37], it would be useful to identify alternative drugs that can remove Pb from human tissues such as cardiomyocytes and neurons with the fewest side-effects maintaining their normal physiological properties. In the current study, the activity of cardiomyocytes was simultaneously monitored during the detection of Pb, which is the major novelty of this study. The combination of Met-lead 1.44 M1 with iPSC-derived cardiomyocytes provides a good platform that could be used to search for candidate anti-Pb drugs (e.g., with effective and specific ability to chelate Pb) while simultaneously monitoring the cardiac toxicity of Pb by measuring beating activity (bpm, beats per minute, Figure 2D,E and Figure 3D,E; Table 1). The development of a Pb-monitoring system that can also sense cardiac activity through its incorporation of a near-infrared genetically encoded calcium ion biosensor (NIR-GECO1) [38] may achieve this aim in the future because the spectrum of NIR-GECO1 does not overlap with that of Met-lead (CFP and YFP).

Our work here also served to introduce a promising animal model for studying Pb poisoning: because the *UAS-Gal4* system can be used to express the genetically encoded Pb biosensor in a tissue-specific manner, the *UAS-Met-lead 1.44 M1* fly strain can be used for further toxicological investigations of Pb-sensitive organs. For example, the influence of Pb on neuronal activity or circuitry can be revealed by monitoring neurons that express Met-lead, such as cholinergic neurons in adult fly brain ex vivo (Appendix A) or in vivo (Figure 4 and Appendix A in single-cell resolution), and larval flies in situ (various regions within different depth/optical sections of the larval CNS, as shown in Figure 5). The use of 3D microscopic imaging of single cells within intact fly (adult brain or larval CNS), together with FRET Pb biosensing, can create an overview of Pb neurotoxicity at a resolution that has not previously been achievable with the help of advanced sample preparation to efficiently preserve the FRET ratio value after fixation and tissue clearing. In the future, this could also be enhanced by using NIR-GECO which combines single-neuronal activity monitoring (the NIR version of GECO can be used to measure Ca transient as an activity assay within a deep tissue region) and Pb sensing via two-photon in vivo FRET [39].

## 5. Conclusions

In this study, we demonstrated that the optimized biosensor Met-lead 1.44 M1 is a useful tool for both Pb detection and observing the activity of Pb-targeted populations, i.e., the cardiovascular and neuronal systems. Real-time observation using Met-lead 1.44 M1 showed that the beating activity of iPSC-derived cardiomyocytes was completely blocked by Pb. The combination of a Pb biosensor with the semi-tissue platform for the cardiovascular system could be utilized to search for workable anti-Pb drugs. Furthermore, two-photon 3D FRET ratio imaging integrated with the Pb biosensor could potentially be used to explore the possible Pb-sensitive neuronal circuitry within the adult brain or larval CNS of the *Drosophila* model in the future. In summary, this study demonstrates the use of versatile cell and animal models for the biosensing of Pb to provide an easily-to-use tool for advanced investigations into lead toxicology.

## Figures and Tables

**Figure 1 biosensors-11-00371-f001:**
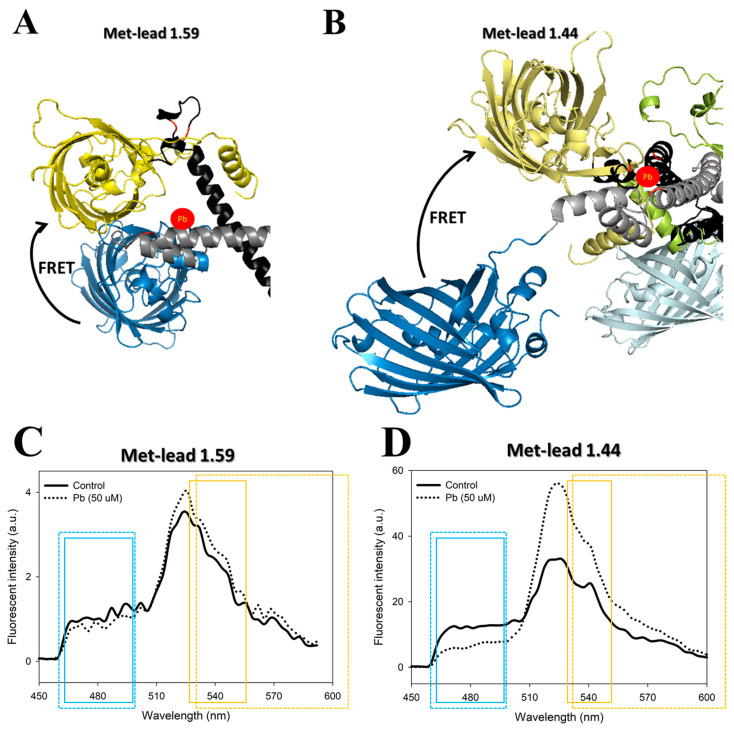
The simulated structure of FRET-based Met-lead Pb sensors. (A,B), The FRET pair of fluorescent proteins within Met-lead ((**A**), version 1.59; (**B**), version 1.44) are depicted: ECFP (blue) and EYFP/Venus (yellow) led to FRET in the presence of the target (Pb, red circles), due to Pb binding by the sensing key residues (three red lines near Pb). This image was adapted from a previous report [26]. (**C**,**D**), Spectral comparisons of donor emission (ECFP) and acceptor emission (EYFP/Venus) between Met-lead 1.59 (**C**) and Met-lead 1.44 (**D**) using spectrum scans under 440 nm laser illumination (excitation source). More detailed information about the excitation and emission ranges of ECFP and EYFP/Venus is provided in Appendix A. The solid (in-cell FRET) and dashed (two-photon FRET) rectangles in C and D indicate the filter settings for the emission ranges of ECFP (blue) or EYFP/Venus (yellow). The spectral data was adapted from a previous report [23].

**Figure 2 biosensors-11-00371-f002:**
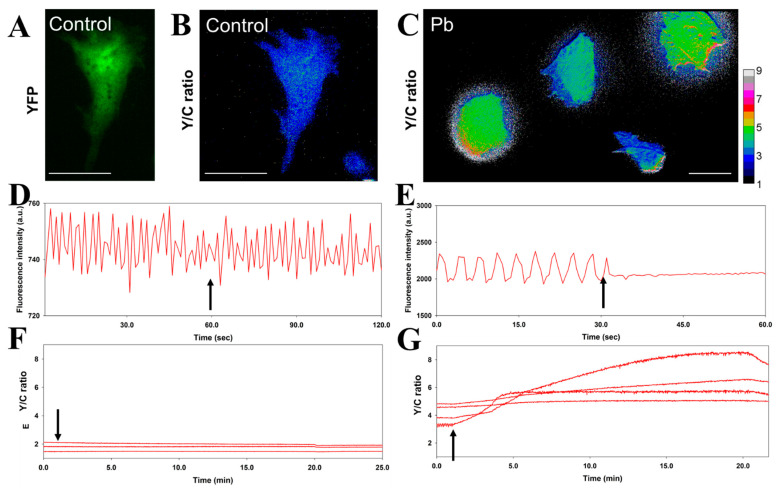
The simultaneous recording of the physiological activity and entry of Pb into iPSC-derived cardiomyocytes. (**A**–**C**), The acceptor (YFP) image (**A**) and the FRET Y/C ratio images (**B**,**C**) of cells expressing Met-lead 1.44 M1 were acquired in control buffer (**A**,**B**) or after Pb treatment (50 μM; (**C**)). The color bar for B and C represents Y/C ratio values from 1 to 9. The scale bars are 40 μm. (**D**,**E**), Within the 25-min recordings, the time-lapse fluctuation of fluorescence signals in the tested cells revealed the beating activity (control in (**D**); Pb in (**E**)). (**F**,**G**), Simultaneously, intracellular Pb sensing (Y/C ratio increase) could be observed in control (**F**) and Pb (**G**) treatments. Buffer without (Control) or with Pb (Pb) was introduced at the time points indicated by arrows.

**Figure 3 biosensors-11-00371-f003:**
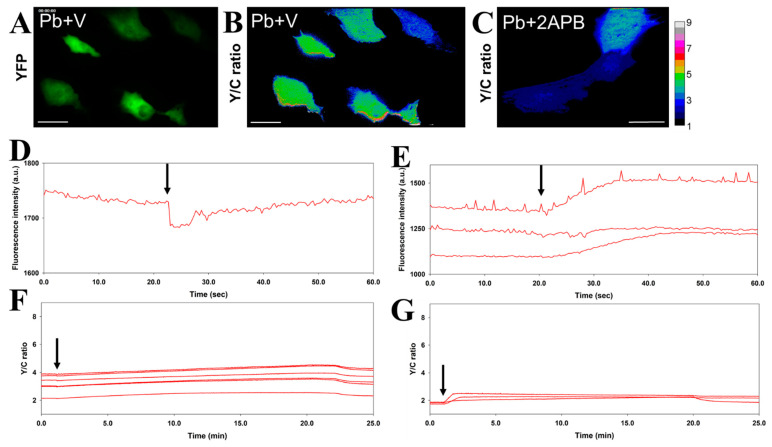
Assessment of the differential effects of Ca^2+^ blockers on iPSC-derived cardiomyocytes by the simultaneous recording of beating activity and Pb sensing. (**A**–**C**), The acceptor (YFP) image (**A**) and the FRET Y/C ratio images (**B**,**C**) of cells expressing Met-lead 1.44 M1 were acquired in Pb buffer with verapamil (Pb + V; (**A**,**B**)) or in Pb buffer with 2-aminoethoxydiphenyl borate (2-APB) (Pb + 2APB; (**C**)). The color bar for B and C represents Y/C ratio values from 1 to 9. The scale bars are 40 μm. (**D**,**E**), During the 25-min recordings, the time-lapse fluctuation in fluorescence signals in the tested cells revealed the beating activity (**D**,**E**). (**F**,**G**), Intracellular Pb sensing can be observed for Pb in the presence of verapamil (**F**) and for Pb in the presence of 2-APB (**G**). Pb (50 μM) was introduced at the time points indicated by arrows.

**Figure 4 biosensors-11-00371-f004:**
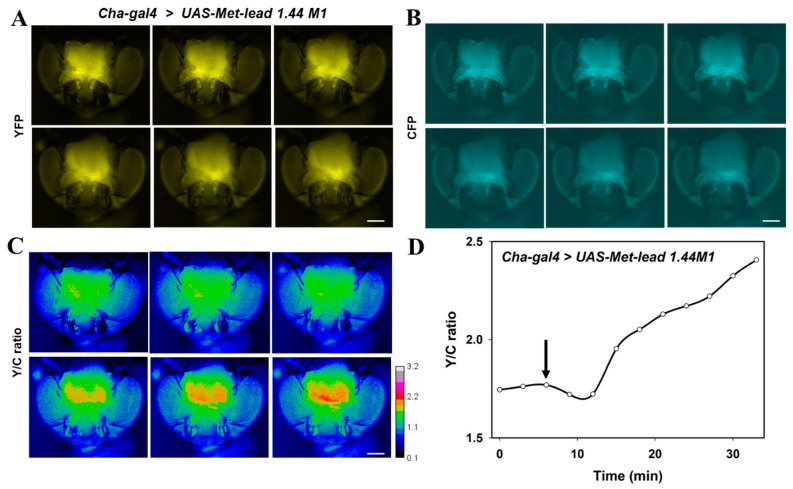
In vivo Pb biosensing in the adult brain of Drosophila (strain Cha-gal4 > UAS-Met-lead 1.44 M1) using FRET ratio live imaging. (**A**–**C**), Representative fluorescence images of YFP (**A**) and CFP (**B**) emission, and the Y/C ratio (YFP/CFP, (**C**)), in the fly brain during a 35-min recording. The color bar in C represents ratio values from 0.1 to 3.2. The scale bars are 100 μm. (**D**), Time-lapse changes in the Y/C ratio value (YFP/CFP) during the in vivo recording. Pb (10 μM) was introduced at the time point indicated by arrows until the end of the recording.

**Figure 5 biosensors-11-00371-f005:**
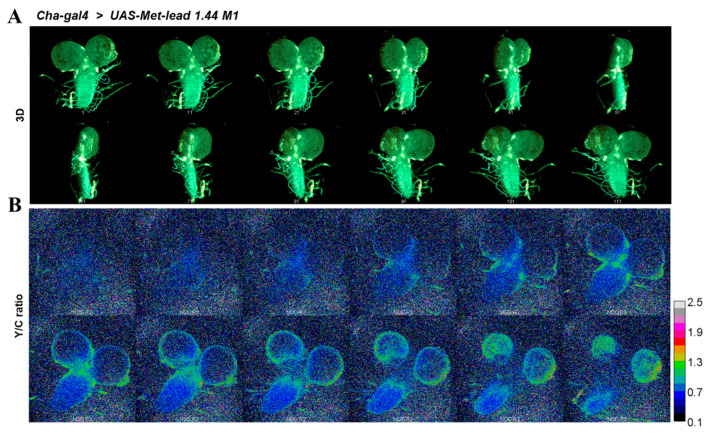
In situ Pb biosensing in the larval CNS of *Drosophila* using high-resolution two-photon FRET ratio imaging. (**A**), A 3D reconstructed (projection) image of the larval CNS taken using a two-photon FRET ratio microscope (41 sections in total) was rotated by 360°. (**B**), Twelve representative sections of Y/C ratio images are shown. The color bar in B represents ratio values from 0.1 to 2.5. The scale bars are 100 μm. (**C**), The emission ratio values of these 3D-sectioned FRET Y/C ratio images are displayed for different depths (slice number) and certain regions were selected, such as the right (○) and left (△) parts of the larval brain hemisphere and the branch (☆) and center (σ) of the spinal cord.

**Table 1 biosensors-11-00371-t001:** Intracellular entry of Pb (FRET ratio changes) and the physiological impact of Pb (beating rate) on iPSC-induced cardiomyocytes. The original data were extracted from Figure 2 and Figure 3. The beating frequency was described in Appendix A.

iPSC Experimental Sets	Delta Pb (FRET Y/C Ratio)	Beating Frequency
Control	NA ^1^ (basal maintained at 1.97 ± 0.15)	24 bpm ^2^ → 24 bpm
Pb (50 μM)	2.06 (4.18 ± 0.32 → 6.24 ± 0.17)	20 bpm → NA
Pb (50 μM) + Verapamil	0.19 (3.06 ± 0.14 → 3.25 ± 0.73)	NA
Pb (50 μM) + 2-APB	0.34 (1.91 ± 0.1 → 2.25 ± 0.47)	20 bpm → 16 bpm

^1^ NA: none detected. ^2^ bpm: beats per minute.

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
