# Peer review of "Versatile Cell and Animal Models for Advanced Investigation of Lead Poisoning"

_biosensors, 2021, doi:10.3390/bios11100371_

Round 1

Reviewer 1 Report

The manuscript written by Yang and Chang, described an interesting FRET sensor composed of fluorescent proteins for in vivo and in vitro detection of Pb. The displayed results may have a potential application in the development of a sensing/imaging kit. However, the study is premature and it is required a deep review of the manuscript. Even so, I am going to name some concerns that in my opinion could improve the article, if addressed.

  1. In a general way, the MS would benefit from a review and rewriting. The abstract, that is the first part once read about the work, is too specific and does not read smoothly while the conclusions are too basic. The discussion part is described superficially, and not all the results observed are discussed, explained, and justified.
  2. The letter F in FRET should refer to “Förster” and not “fluorescence” (Abstract and line 62).
  3. Line 18: reads: “Pb was effectively sensed in two living models producing Met-led 1.44 M1”. This sentence doesn’t make sense.
  4. References to be added in line 46 and line 65.
  5. Please, expand the abbreviatures: GEFP (line 58), BFP and GFP (line 62), CFP and YFP (line 63), DRET (line 161).
  6. The concept described in the following two sentences is quite similar and it is redundant and repetitive. Please rewrite this part of the introduction. “Fluorescent indicators in combination with suitable readout platforms can be used to monitor the presence of specific compounds or elements within living systems or in their environment.” AND “Strategies for the qualitative and quantitative sensing of targets by GEFP biosensors can include the combination of a bioreceptor (a selected sensing domain) with a signal transducer, which transforms target recognition events into fluorescent signals.”
  7. A complete comparison of the previously developed biosensor Met-lead 1.59 and the new one Met-lead 44 M1 should be included in the discussion part to highlight the new features of the 1.44M1. The authors say in the introduction that the structural difference between them is that a linker spacer was added in Met-lead 1.44 M1 biosensor. Then, I guess that now the donor and acceptor pairs are farther. Taking into account that the FRET sensing ability is strongly affected by the D/A pair distance, how can the authors explain the better sensing efficiency when D and A are located farther?
  8. A figure including the donor emission and acceptor absorption of the FRET pairs should be included or in the MS or the ESI.
  9. Scale bars should be added to the microscope images (figure 2, 3, 4, 5). Also, all the microscope images would benefit from a DIC image.
  10. Figure caption 2: The color bar for B and C represents ratio values from 1 to 9, there is non-color bar in B.
  11. The authors used this expression repetitively along the MS: “The fluorescence and FRET ratio images”, both images are fluorescent ones. I suggest calling it like images of the donor and FRET channel or images of the donor and FRET emission.
  12. How do you calculate the FRET ratio images? What is the Ratio Plus plugin, it is an addition of both images?
  13. A material section needs to be added with the material employed during the work and specifying the FP used. Did you synthesized the final biosensor or is commercial?
  14. The setup for FRET imaging (excitation source/filters) is not explained along the MS. Otherwise, I was wondering if the author uses the correct expressions and images along the MS. In the Data Analysis part, it is written that YFP channel and CFP channel, which means donor (excitation and emission of the donor) and acceptor (excitation and emission of the acceptor). However, to calculate the FRET ratio is required the ratio between FRET channel (donor excitation and acceptor emission) and the donor emission. Please, check in other publications about FRET how FRET ratio is calculated (I may suggest publications of Prof. Niko Hildebrandt).
  15. Some spelling errors: line 18 Met-led 1.44 M1, Figure 1: Met-lead 59

Author Response

Point 1: In a general way, the MS would benefit from a review and rewriting. The abstract, that is the first part once read about the work, is too specific and does not read smoothly while the conclusions are too basic. The discussion part is described superficially, and not all the results observed are discussed, explained, and justified.

Response 1: We thank Reviewer’s comments. We have rewritten the Discussion including: Line 240-254 (also for Response 7) on structure & comparison of 1.59 and 1.44 (Figure 1);

Line 255-265 (also for Response 14) about the ratio imaging (filter setup for FRET; with additional Figure S1, Figure 1C and 1D) and the future possibility of using portable device with titration and FRET calculation in practical environmental detection (two new refs added);

Line 266-275 (also for Response 8 of Reviewer 2) highlight the novelty of Met-lead can be a powerful drug screen platform to be used (Figure 2 and 3); and

Line 287-295 with more detail about the in vivo/in situ animal imaging (Figure 4 and 5).

Due to the lack of crystal structure on PbrR, the central heart of Met-leads, the structural knowledge on our Pb biosensors relays on prediction/simulation which still needs to be confirmed further. Fortunately, PbrR belongs to a superfamily MerR and the crystallography of some MerR member has been resolved [33]. The use of homology modelling (taking the structural information of MerR as reference to calculate that of PbrR) to display PbrR [23] and Met-leads (Figure 1A and 1B) [23,26] could be nearly to the real status of them. The distance between donor and acceptor inside Met-lead 1.44 seems to be farther than that inside Met-lead 1.59. Whether it is the major reason for higher FRET efficiency of Met-lead 1.44 than that of Met-lead 1.59 remains to be verified by solid crystallography data in the future. While we propose that there could be with more space between FRET pair proteins inside Met-lead 1.44 than that within Met-lead 1.59, it could generate various kinds of flexible conformation for FRET (thus producing wilder ranges of FRET ratio values), maybe this can explain the high dynamic ranges and sensitivity of Met-lead 1.44. In addition, the linker version (1.44 M1) may generate additional space between FRET pair proteins to drive FRET closer (lower ratio value at resting [23]). (Line 240-254)

The experimental data presented in this study demonstrate the qualitative analysis of Y/C ratio results similar as previous reports [15]. The increased FRET ratio can be confirmed as Pb signals due to two facts: 1) The additional information (Figure S1; Figure 1C and 1D) indicates the filter setup within W-View module can effectively separate the emission of CFP and that of YFP and thus the cross-over between CFP and YFP can be prevented. 2) The increased ratio values have been proved to be specifically from Pb using the Pb chelator (TPEN) as previous report ([23]) stated, similar to the Ca biosensor ([15]). For further practical quantitative monitoring of Pb to be applied in many fields like environmental detections, the titration of Pb (various concentrations, e.g. from as low as 1 nM to 50 μM) through the optimized Met-lead biosensor [23] can be integrated with new developed portable/smartphone device [34], as well as the precise calculation of FRET [35]. (Line 255-265)

Compared with our previous study, the activity of cardiomyocytes was further monitored simultaneously during the detection of Pb. Because concerns exist about the use of chelating agents … Through additional analysis on the beating parameters (bpm, beats per minute, as shown in Table 1) of cardiomyocytes, their physiological condition can be dedicatedly monitored during the screening of anti-Pb drug (checking whether candidates have good chelating ability also are with the fewest side-effects at the same time). Thus, the combination of … (Line 266-275)

… cholinergic neurons in adult in vivo (Figure 4) or larval flies in situ (various regions within different depth (section) of the larval CNS as shown in Figure 5). The use of 3D microscopic imaging of single cells within intact fly larva, together with FRET Pb biosensing, can create an overview of Pb neurotoxicity at a resolution that has not previously been achievable with the help of advanced sample preparation to efficiently preserve the FRET ratio value after fixation and tissue clearing. In the future, this could also be enhanced by using NIR-GECO which combines the single-neuronal activity monitoring (the NIR version of GECO can be used to measure Ca transient as activity assay within deep tissue region) and the Pb sensing under two-photon in vivo FRET [39]. (Line 287-295)

In addition, we’ve also rewritten the contents of Conclusion as:

In summary, this study demonstrates the use of versatile cell and animal models for the biosensing of Pb to provide an easily-to-use tool for advanced investigations into lead toxicology. Such biosensing tool for Pb detections both for the major Pb-sensitive targets, i.e. neuronal and cardiovascular systems will be very important to be applied in the future.

The added References are:

[33] Changela, A.; Chen, K.; Xue, Y.; Holschen, J.; Outten, C.E.; O'Halloran, T.V.; Mondragón, A. Molecular basis of metal-ion selectivity and zeptomolar sensitivity by CueR. Science. 2003, 301, 1383-1387.

[34] Chang, T.J.; Lai, W.Q.; Chang, Y.F.; Wang, C.L.; Yang, D.M. Development and optimization of heavy metal lead biosensors in biomedical and environmental applications. J. Chin. Med. Assoc. 2021, 84, 745-753.

[35] Algar, W.R.; Hildebrandt, N.; Vogel, S.S.; Medintz, I.L. FRET as a biomolecular research tool - understanding its potential while avoiding pitfalls. Nat. Methods. 2019, 16, 815-829.

The modified Abstract is shown in Response 1 of Reviewer 2.

Point 2: The letter F in FRET should refer to “Förster” and not “fluorescence” (Abstract and line 62).

Response 2: We agree the Reviewer’s comments. We have modified the sentences as:

… We applied the optimized Förster ... (Line 13)

… was developed to detect Förster resonance energy transfer (FRET) signal events …. (Line 65)

Point 3: Line 18: reads: “Pb was effectively sensed in two living models producing Met-led 1.44 M1”. This sentence doesn’t make sense.

Response 3: We have modified the sentence as:

Pb was effectively sensed in two living models genetically encoding and expressing Met-lead 1.44 M1. (Line 18)

Point 4: References to be added in line 46 and line 65.

Response 4: We added the references as:

… harmful effects for patients with ages ranging from 4 to 85 years old [9]. (Line 49)

… in a genetically controlled manner without the need to stain tissues [13]. (Line 70)

  • [9] Central News Agency of. Chinese medicine doctors, dealer detained in lead poisoning case. Taipei News 2020, 2020/08/07. https://www.taiwannews.com.tw/en/news/3982844
  • [13] Carter, K.P.; Young, A.M.; Palmer, A.E. Fluorescent sensors for measuring metal ions in living systems. Chem. Rev. 2014, 114, 4564–4601.

Point 5: Please, expand the abbreviatures: GEFP (line 58), BFP and GFP (line 62), CFP and YFP (line 63), DRET (line 161).

Response 5: We have modified the sentences as:

… alternatively, they can be genetically encoded (GE), as with fluorescent protein (FP) bio-sensors (i.e. GEFP biosensors) that can … (Line 57-58)

… between two pairs of FPs (blue fluorescent protein, BFP with green fluorescent protein, GFP, or cyan fluorescent protein, CFP with yellow fluorescent protein, YFP) … (Line 66-67)

In Figure 5: A, …  using a two-photon FRET ratio microscope (41 sections in total) … (Line 226)

Point 6: The concept described in the following two sentences is quite similar and it is redundant and repetitive. Please rewrite this part of the introduction. “Fluorescent indicators in combination with suitable readout platforms can be used to monitor the presence of specific compounds or elements within living systems or in their environment.” AND “Strategies for the qualitative and quantitative sensing of targets by GEFP biosensors can include the combination of a bioreceptor (a selected sensing domain) with a signal transducer, which transforms target recognition events into fluorescent signals.”

Response 6: We thank Reviewer’s comments. We think the original description may mislead the audience, so we modified these sentences as:

Fluorescent indicators can be in chemical form, which are preferentially applied for environmental sensing; alternatively, they can be genetically encoded (GE), as with fluorescent protein (FP) biosensors (i.e. GEFP biosensors) that can be expressed intracellularly to sense specific targets. In combination with suitable readout platforms, both kinds of fluorescent indicators can be applied to monitor the presence of specific compounds or elements wherever within living systems or in the environment [13,14]. Strategies for the qualitative and quantitative sensing of targets by GEFP biosensors can include the combination of a bioreceptor (a selected sensing domain) with a signal transducer, which transforms target recognition events into fluorescent signals. (Line 56-64)

Point 7: A complete comparison of the previously developed biosensor Met-lead 1.59 and the new one Met-lead 44 M1 should be included in the discussion part to highlight the new features of the 1.44M1. The authors say in the introduction that the structural difference between them is that a linker spacer was added in Met-lead 1.44 M1 biosensor. Then, I guess that now the donor and acceptor pairs are farther. Taking into account that the FRET sensing ability is strongly affected by the D/A pair distance, how can the authors explain the better sensing efficiency when D and A are located farther?

Response 7: We agree the Reviewer’s comments. We have added the comparisons between Met-lead 1.59 and Met-lead 44 M1 as shown in Response 1 (Line 241-255).

Point 8: A figure including the donor emission and acceptor absorption of the FRET pairs should be included or in the MS or the ESI.

Response 8: We have added the donor emission and acceptor absorption of the FRET pairs within data of Figure 2, 3, and 5 in additional ESI (Additional Figures) as:

Figure S1. Original data for the assessment of the physiological activity and entry of Pb on iPSC-derived cardiomyocytes (Figure 2). Representative sections of control (Control; A) or in Pb buffer (P; B) in YFP and CFP channels are shown.

Figure S2. Original data for the assessment of the differential effects of Ca2+ blockers on iPSC-derived cardiomyocytes (Figure 3). Representative sections of Pb buffer with verapamil (P+V; A) or in Pb buffer with 2-aminoethoxydiphenyl borate (2-APB) (P+2APB; B) in YFP and CFP channels are shown.

Figure S3. Original data of in vivo Pb biosensing in the larval CNS of Drosophila for Figure 5. Representative sections of larval CNS in YFP and CFP channels are shown in A and B respectively. The scale bar is 100 μm.

Point 9: Scale bars should be added to the microscope images (figure 2, 3, 4, 5). Also, all the microscope images would benefit from a DIC image.

Response 9: We have added scale bars in Figure 2, 3, 4, 5. There’s no DIC in our FRET setup.

Point 10: Figure caption 2: The color bar for B and C represents ratio values from 1 to 9, there is non-color bar in B.

Response 10: We have modified the Figures 2 and 3 and the sentences as:

The color bar for B and C represents ratio values from 1 to 9. (Line 184)

The color bar for B and C represents ratio values from 1 to 9. (Line 193)

Point 11: The authors used this expression repetitively along the MS: “The fluorescence and FRET ratio images”, both images are fluorescent ones. I suggest calling it like images of the donor and FRET channel or images of the donor and FRET emission.

Response 11: Thank Reviewer’s suggestions. We have modified the descriptions as:

… A-C, The acceptor (YFP) image (A) and the FRET ratio images (B and C) of cells …  (Line 183)

… A-C, The acceptor (YFP) image (A) and the FRET ratio images (B and C) of cells …  (Line 191)

Point 12: How do you calculate the FRET ratio images? What is the Ratio Plus plugin, it is an addition of both images?

Response 12: “Ratio Plus plugin”, is a java written by Dr. Paulo J. Magalhães, University of Padua, Italy for the software ImageJ. Detail information can be found as:

https://imagej.nih.gov/ij/plugins/ratio-plus.html

Practically, the procedure is like the following steps:

More information is as:

Source:    Ratio_Plus.java (https://imagej.nih.gov/ij/plugins/download/Ratio_Plus.java)

Installation:     Download Ratio_Plus.class (https://imagej.nih.gov/ij/plugins/download/Ratio_Plus.class) to the plugins folder, or subfolder, and restart ImageJ.

Description:    

This plugin calculates the ratio between two images (single frames or stacks),

as used in Fura-2 experiments, for example. The code -- heavily annotated to help

beginners like myself explore the world of ImageJ -- is largely based on Image

Calculator Plus. Paulo Magalhães, 10dec03.

The plugin requires two images of the same width and height,

and of the same type (8-, 16-, or 32-bit); the images must be:

  1. i) two single images;
  2. ii) two stacks with the same number of frames; or

  iii) a stack as a first image and a single frame as a second - in this case,

       the single frame will be applied throughtout the complete stack.

The resulting image (in 32-bit) is calculated as follows:

    intR = (intA - bkgA) / (int B - bkgB) * MF

where intA, intB and intR are the intensities of the first, second and ratio images,

respectively; bkgA and bkgB are constant background values (entered by the user)

for the first and second images, respectively; MF is an arbitrary multiplication factor.

If a pixel value becomes negative after background subtraction, it is set to zero

before ratio calculation.

Point 13: A material section needs to be added with the material employed during the work and specifying the FP used. Did you synthesize the final biosensor or is commercial?

Response 13: Thank Reviewer’s comments. We did not synthesize or but the FP. Basically, certain FP worked in this study came out (genetically-encoded) from specific gene sequence and presented through the method transfection to the cells or the transgenic way on flies.

Point 14: The setup for FRET imaging (excitation source/filters) is not explained along the MS. Otherwise, I was wondering if the author uses the correct expressions and images along the MS. In the Data Analysis part, it is written that YFP channel and CFP channel, which means donor (excitation and emission of the donor) and acceptor (excitation and emission of the acceptor). However, to calculate the FRET ratio is required the ratio between FRET channel (donor excitation and acceptor emission) and the donor emission. Please, check in other publications about FRET how FRET ratio is calculated (I may suggest publications of Prof. Niko Hildebrandt).

Response 14: Thank Reviewer’s suggestions. We provided Figure S4 for the explanation of why using 440 nm laser as excitation source and why using the specific ranges (as shown in additional Figure 1C/1D) for the filter setup in this study to avoid excitation crossover (e.g. for ECFP: 483/32 nm on in-cell FRET, blue solid line in Figure 1C/1D; 460–500 nm on two-photon FRET, blue dashed line in Figure 1C/1D; for EYFP/Venus: 542/27 nm on in-cell FRET, yellow solid line in Figure 1C/1D; 530–630 nm on two-photon FRET, yellow dashed line in Figure 1C/1D).

Figure S1. Spectral analysis of the FRET pairs (ECFP and EYFP/Venus) for the considerations of related filter setup. The spectral relationship between donor (ECFP in light blue) excitation (dash line)/emission (solid line) with acceptor (EYFP/Veuns in yellow) excitation (dash line)/ emission (solid line) under the emission and excitation spectrum scan of these FPs. The dark blue rectangle space indicates the excitation laser source for ECFP (435 nm). The dashed rectangle spaces indicate the emission ranges of ECFP (in dark cyan) and those of EYFP/Venus (in dark yellow).

Figure 1. …. C, D, The spectral comparisons between Met-lead 1.59 (C) or Met-lead 1.44 M1 (D) on donor emission (ECFP) and acceptor excitation (EYFP/Venus) in the emission spectrum scan. The excitation laser source is 440 nm (detail information in Figure S1). The solid (in-cell FRET) and dashed (two-photon FRET) rectangle spaces shown in C and D indicate the filter settings for the emission ranges of ECFP (blue) or EYFP/Venus (yellow). (Line 102-106)

And we agree Reviewer’s suggestion. We add the sentences in the Discussion part as shown in Response 1 (Line 256-266).

Point 15: Some spelling errors: line 18 Met-led 1.44 M1, Figure 1: Met-lead 59.

Response 15: Thank Reviewer’s suggestions. We have corrected the spelling errors as:

Pb was effectively sensed in two living models genetically encoding and expressing Met-lead 1.44 M1 (line 19)

Reviewer 2 Report

1) Abstract is misleading by containing summary of results not presented in the submitted manuscript. This need to be revised to accurately represent the submitted work. For example, no limit of detection measurements were performed in this work.
2) Presented cardiomyocytes beating data/analysis is wrong.  Fig. 2D and E show beating of ~5bpm or 0.08Hz, which is very slow (but probable). For reference and for beating frequency analysis see doi:10.3390/photonics5040039 .
3) No experimental details regarding the Pb exposure are provided (when and for how long). Verapamil and 2-APB pretreatment details are missing.
4) For Figs. 2 and 3, the location of line plot data needs to be clearly identified in the images. What is the reason for the significant noise and drift in your fluorescence measurements?
5) For physiologically relevant sensing your system has a concentration issue: Your biosensor has LOD of 10nM and dynamic range of 10mM (previously reported), but Pb treatment reported was 50mM. As noted in your previous works, this is orders of magnitude higher than the acceptable exposure limit. How translational is pathology at 1000x concentration?
6) Details of Pb exposure for in vivo and ex vivo experiments are missing.
7) As previously reported, two-photon experiments should be called ex vivo, in place of in vivo.
8) The novelty of this work over previous is not presented in discussion or conclusion.

Author Response

Point 1: Abstract is misleading by containing summary of results not presented in the submitted manuscript. This need to be revised to accurately represent the submitted work. For example, no limit of detection measurements were performed in this work.

Response 1: We have removed the result-unrelated contents (… with a limit of detection of 10 nM (2 ppb) …) and modified the Abstract as (also for Response 1 of Reviewer 1):

The heavy metal lead (Pb) can irreversibly damage the human nervous system. To help understand Pb-induced damage, we have developed practical applications for genetically encoded Pb biosensors in cardiac cells and insect central nervous tissue. We applied the optimized Förster resonance energy transfer (FRET)-based Pb biosensor Met-lead 1.44 M1 to two living systems to monitor the concentration of Pb: induced pluripotent stem cell (iPSC)-derived cardiomyocytes as a semi-tissue platform, and Drosophila melanogaster fruit flies as an in vivo animal model. Different FRET imaging modalities were used to obtain FRET signals, which represented the presence of Pb in the tested samples in different spatial dimensions. Pb was effectively sensed in two living models genetically encoding and expressing Met-lead 1.44 M1. In iPSC-derived cardiomyocytes, the relationship between beating rate determined from the fluctuation of fluorescent signals and the concentrations of Pb represented by the FRET emission ratio values of Met-lead 1.44 M1 demonstrated the potential of this fluorescence biosensor system for anti-Pb drug screening. In the Drosophila model, Pb was detected within the adult brain or larval central nervous system using fast epifluorescence and high-resolution two-photon 3D FRET ratio image systems. The optimized Pb biosensor together with FRET microscopy can be used for specific applications to detect Pb with a limit of detection of 10 nM (2 ppb). We believe that this integrated Pb biosensor system can be applied to the prevention of Pb poisoning.

Point 2: Presented cardiomyocytes beating data/analysis is wrong.  Fig. 2D and E show beating of ~5bpm or 0.08Hz, which is very slow (but probable). For reference and for beating frequency analysis see doi:10.3390/photonics5040039.

Response 2: We thank Reviewer’s comments. The time unit of y axis of Figures 2D, 2E, 3D, 3E was mislabelled to minutes. We correct (min) to (sec) in them and recalculate the beating frequency as shown in Table 1. Since our microscope is not equipped with DIC image set-up, we simply detect the signal of CFP or YFP over time. Movement of object (beating of cardiomyocytes) resulting in focus change upon contraction would lead to intensity change detected by camera.

Point 3: No experimental details regarding the Pb exposure are provided (when and for how long). Verapamil and 2-APB pretreatment details are missing.

Response 3: We have added the information about the pretreatment details as:

… Pb was introduced at the time points indicated by arrows as shown in Figures. (Line 117)

… Pb or buffer without Pb (control) was introduced at the time points indicated by arrows. (Line 187)

… Pb was introduced at the time points indicated by arrows. (Line 196)

… Verapamil (final concentration 2 μM) or 2-APB (final concentration 50 μM) were immersed into the buffer containing 2 mM Ca 10 min before the FRET experiments with Pb. (Line 118-120)

Point 4: For Figs. 2 and 3, the location of line plot data needs to be clearly identified in the images. What is the reason for the significant noise and drift in your fluorescence measurements?

Response 4: We thank Reviewer’s comments. We showed the intensity information from single FP in both Figure 2D, 2E and Figure 3D, 3E to display the beating activity of cardiomyocytes. Signals of FPs are generally easy to be disturbed by drug treatment leading to change of micro-environment for cells, which may further slightly alter the cell shape and result in small drift of basal intensity (Figure 3D, 3E). However, all the small drift of signals can be cancelled out by ratiometric imaging shown in Figure 3F and 3G, demonstrating ratiometric based measurements is superior to intensity-based indicators.

Point 5: For physiologically relevant sensing your system has a concentration issue: Your biosensor has LOD of 10nM and dynamic range of 10mM (previously reported), but Pb treatment reported was 50mM. As noted in your previous works, this is orders of magnitude higher than the acceptable exposure limit. How translational is pathology at 1000x concentration?

Response 5: We thank Reviewer’s comments. As our previous paper found that Pb may enter nto living cells through non-selective Ca channels and therefore the existence of Ca (2 mM, general physiological condition) could protect (compete with Pb each other) the entry of Pb. Therefore, it is true that immersing with 50 or 10 μM Pb doesn’t mean the sensing environment (within living cells) equals to the same concentration of Pb.

That is why we must improve the previous version which can’t even significantly monitor the Pb under the existence of Ca (no Ca, no beating). Yes, the DR of optimized Pb biosensor is 10 μM.

Point 6: Details of Pb exposure for in vivo and ex vivo experiments are missing.

Response 6: We thank Reviewer’s comments.

… Pb (50 μM) was introduced at the time points indicated by arrows as shown in Figures. (Line 117)

… Pb (10 μM) was introduced at the time point indicated by arrows. (Line 215)

… Pb (10 μM) was immerse within buffer before the whole larval was fixed and fixated. (Line 230)

Point 7: As previously reported, two-photon experiments should be called ex vivo, in place of in vivo.

Response 7: We thank Reviewer’s suggestion. However, the sample preparation is somehow different from previous paper. Instead of taking out the larval CNS (ex vivo, previous study), the whole larval was immersed in the clear reagent and the following fixation for further FRET imaging. Maybe in situ is better expression. If the Reviewer agree, we modified the in vivo as in situ:

… For in situ (high-resolution, two-photon illumination) FRET biosensing … (Line 131)

Figure 5. In situ Pb biosensing in the larval CNS of … (Line 228)

Point 8: The novelty of this work over previous is not presented in discussion or conclusion.

Response 8: We have modified the Conclusion as shown in Response 1 of Reviewer 1.

Round 2

Reviewer 1 Report

The authors have done a great effort to address all my concerns and the MS is significantly improved. However, there are still some points that need to be clarified.  

  • I have still some concerns about the synthesis of the sensor. Maybe I didn’t ask it correctly the first time. The authors say in line 84 that “to improve the sensitivity of Met-lead molecules in different ways, including adjusting the lengths of the Pb-sensing domains and inserting a repeat sequence (linker) in the center of the Pb-sensing domains”, however, in the present MS the structural difference of both Met.lead 1.59 and 1.44 and especially how the author can control if there is a linker between both proteins or whatever. It has been highlighted that is difficult to know the exact configuration. However, a brief text referring to the procedure of the formation of Met-LEAD 1.44 sensor should be included (even if it has been described before). Also, the ratio between both FP inside the biosensor should be calculated.
  • Some control experiments are missed, especially those to corroborate that the emission of YFP is effectively due to FRET and not to the direct YFP excitation. For example, obtaining the images of a sample with just YFP (without CFP) under exactly the same conditions used for FRET.
  • Scale bars are still missed in some figures, i.e, Fig 2A, Fig 3A, Fig 4, Fig S2, S3, S4
  • The authors have a recent work https://doi.org/10.1016/j.bios.2020.112571 in which they explained the production of different Met-Lead sensors and already applied the 1.44 M1 for detection of Pb in plants and animals (e.g., fruit fly brain neurons). Now I don’t see the novelty of the described system in comparison to the previous works of the authors. I suggest the authors see what they have already published and remarked and highlight all the new things in comparison to their previous work.

Author Response

Response 1: We thank Reviewer’s comments. Basically, the living cells are the sole machine to synthesis most of the genetically-encoded FRET-based biosensor proteins including Met-leads through the gene map designed by the developers [15-23,26]. For the synthesis of the sensor and the procedure of the formation of Met-lead 1.44 sensor, we’ve described the “transfection” method in the manuscript on line 115, and the “transgenic” model in the same concept with tissue-specific benefit. We further added a description in line 69 as: (the bio-synthesis of FRET-based biosensor will proceed through the procedure of DNA transfer and replication, mRNA transcription, and protein translation inside living cells containing the biosensor gene). A successful FRET-based sensing can be demonstrated from previous reports by the increase of Y/C ratio values inside the biosensor-transfected cells upon the existence of target under a FRET ratio imaging under the donor (CFP)-specific excitation (440 nm) to get the emission signals of donor (CFP) and those of acceptor (YFP) [15-23,26]. These data all indicate that the real FRET events (the emission of YFP be excited by the emission of CFP) seems already happened in absence of sensing target, e.g. as shown in left (solid line) and middle left (dash line) of the figure. The existence of sensing target can further enhance such FRET events (the fluorescent signals of YFP increased and those of CFP decrease). In this way, without getting the real YFP signals (under a YFP-specific excitation), a titration (middle right of the figure) combined with a suitable equation can further establish a quantitative calculation of sensing target (right of the figure) [Nagai  et al., 2004; https://doi.org/10.1073/pnas.0400417101]. Although using such increased Y/C ratio values recording might be too simple compared to the classical FRET calculations from original theory established, it is relatively convenient for practical real-time displaying the existence of sensing target only through the donor-specific excitation while getting the emission of both donor and acceptor simultaneously [15-23,26]. Thus, the information of YFP-excited signals might not be essential during this kind of biosensing.

The original motivation of adding linker is to remove the dimer/multimer property of Met-lead [26], and we surprisingly found that the M1 of Met-lead had lower basal ratio than non-M1 form did (could be the result of relatively low energy transfer, i.e. longer distance between YFP and CFP). This could be a clue to explain that M1 can “generate additional space between FRET pair proteins” (line 252). We don’t really know how this happened about the structural changes, although these Met-lead proteins (both 1.59, 1.44 and their M1 versions) seem to be functional express (synthesis) within living cells. If the crystallography data are available to show the real structure of Met-leads in the future, enough information will give the direct answer.

Response 2: We agree the Reviewer’s comments. We have added the scale bars.

Response 3: We thank Reviewer’s suggestions. We’ve added two additional Figure S5 and S6 and modified the Discussion and Conclusion to show the novelty of this work as:

Discussion

Because concerns exist about the clinical use of chelating agents such as dimercaprol (British anti-Lewisite, BAL) and calcium disodium ethylenediaminetetraacetic acid (CaEDTA) to treat Pb poisoning [36,37], it would be … such as cardiomyocytes and neurons with the fewest side-effects maintaining their normal physiological properties. The activity of cardiomyocytes was simultaneously monitored during the detection of Pb which is the major novelty of this study. Such combination of Met-lead 1.44 M1 with iPSC-derived cardiomyocytes provides a good platform with which to search for candidate anti-Pb drugs (e.g. with effective and specific ability to chelate Pb) while monitoring the cardiac toxicity of Pb by measuring beating activity (bpm, beats per minute, Figure 2D, E; Figure 3D, E; Table 1) at the same time. … (NIR-GECO1) [35] may achieve this aim in the future, …

Our work here also served to … cholinergic neurons in adult ex vivo (Figure S5), in vivo (Figure 4; Figure S6 in single-cell resolution, 10 μM Pb with ionomycin) or larval flies in situ (various regions within different depth/optical sections of the larval CNS as shown in Figure 5). The use of 3D … achievable with the help of advanced sample preparation to efficiently preserve the FRET ratio value after fixation and tissue clearing. …

Conclusions

In this study, the optimized biosensor Met-lead 1.44 M1 was well demonstrated as a useful tool for both Pb detections and observing the activity of Pb-targeted populations, i.e. the cardiovascular and the neuronal systems. Real-time observation using Met-lead 1.44 M1 shows that beating activity of iPSC-derived cardiomyocytes were completely blocked by Pb. The combination of Pb biosensor with the semi-tissue platform for cardiovascular system can be applied for the search of workable anti-Pb drugs. Two-photon 3D FRET ratio imaging integrated with the Pb biosensor further provide future hope to explore the possible Pb-sensitive neuronal circuitry within the adult brain or larval CNS of Drosophila model. In summary, this study demonstrates the use of versatile cell and animal models for the biosensing of Pb to provide an easily-to-use tool for advanced investigations into lead toxicology.

Reviewer 2 Report

Authors included good revisions. However, there are still some concerns over the BPM analysis.  

- No details of BPM analysis are provided.  If no beating was detected after Pb treatment in Fig. 2G, then your system is subject to significant noise, as evident from line plots from 10 to 20 mins.  This would require significant denoising, which is not described. At the same time, in Fig. 3E only 1 out of 3 line plots demonstrate visual beating. Your BPM analysis would have to be very robust to manage your experimental conditions and full details need to be included.

Your revisions need minor proofreading:

-line 21, rephrase "verapamil (2um) can't reverse"

- line 60, delete "wherever"

-line 240, "relays" to "relies"

- line 243, "wilder" to "wider"

Author Response

Response 1: We thank Reviewer’s comments. We added detail about the BPM analysis as:

Figure S4. The analysis of beating frequency from the fluorescent recording of iPSC-derived cardiomyocytes. Generally, we put the ROI close to the edge of the cells as shown in A. Contraction of the cells causing the repetitive movement of object shown in ROI (A2) would change the total intensity of ROI as plotted in B.

Table 1. Intracellular entry of Pb (FRET ratio changes) and the physiological impact of Pb (beating rate) on iPSC-induced cardio-180 myocytes. The original data were extracted from Figures 2 and 3. The beating frequency was described in Figure S4.

Response 2: We thank Reviewer’s comments. We have corrected these errors as:

Line 21: verapamil (2 μM) can’t reverse the ceasing of beating …

Line 60: ”wherever” deleted …

Line 240: relies

Line: 248: wider

Round 3

Reviewer 1 Report

Thanks to the authors to answer appropriately all my comments.

1- The authors have added a justification of the FRET setup that is convincing. However, they said in line 260 "Our results of qualitative analysis of Y/C ratios based on the current experimental 260 data are similar to those of previous reports." I was not able to find this qualitative analysis along the MS. I encourage the authors to add the Y/C ratio since it is important for developing FRET-based sensors.

2- Some typo errors

Line 58 “i.e.,”

Line 255 “higher”

Line 308 “,.i.e.,”

Line 212 “show”

Author Response

Point 1: 1- The authors have added a justification of the FRET setup that is convincing. However, they said in line 260 "Our results of qualitative analysis of Y/C ratios based on the current experimental 260 data are similar to those of previous reports." I was not able to find this qualitative analysis along the MS. I encourage the authors to add the Y/C ratio since it is important for developing FRET-based sensors.

Response 1: We agree Reviewer’s suggestions. We unified all the expressions “Emission ratio” or “Ratio” as “Y/C ratio” to let the audience clearly understand the meaning of qualitative analysis in this study.

Point 2: Some typo errors Line 58 “i.e.,” Line 255 “higher” Line 308 “,.i.e.,” Line 212 “show”

Response 2: We thank Reviewer’s comments. We have corrected all the typo errors.
